# Convolutional Neural Network Classifies Pathological Voice Change in Laryngeal Cancer with High Accuracy

**DOI:** 10.3390/jcm9113415

**Published:** 2020-10-25

**Authors:** HyunBum Kim, Juhyeong Jeon, Yeon Jae Han, YoungHoon Joo, Jonghwan Lee, Seungchul Lee, Sun Im

**Affiliations:** 1Department of Otolaryngology-Head and Neck Surgery, Bucheon St. Mary’s Hospital, College of Medicine, The Catholic University of Korea, Seoul 06591, Korea; goldgold11@hanmail.net (H.K.); joodoct@catholic.ac.kr (Y.J.); 2Department of Mechanical Engineering, Pohang University of Science and Technology (POSTECH), Pohang 37673, Korea; jjeon@postech.ac.kr (J.J.); leejhd@postech.ac.kr (J.L.); 3Department of Rehabilitation Medicine, Bucheon St. Mary’s Hospital, College of Medicine, The Catholic University of Korea, Seoul 06591, Korea; duswohan@gmail.com; 4Graduate School of Artificial Intelligence, Pohang University of Science and Technology (POSTECH), Pohang 37673, Korea

**Keywords:** voice change, larynx cancer, machine learning, deep learning, voice pathology classification

## Abstract

Voice changes may be the earliest signs in laryngeal cancer. We investigated whether automated voice signal analysis can be used to distinguish patients with laryngeal cancer from healthy subjects. We extracted features using the software package for speech analysis in phonetics (PRAAT) and calculated the Mel-frequency cepstral coefficients (MFCCs) from voice samples of a vowel sound of /a:/. The proposed method was tested with six algorithms: support vector machine (SVM), extreme gradient boosting (XGBoost), light gradient boosted machine (LGBM), artificial neural network (ANN), one-dimensional convolutional neural network (1D-CNN) and two-dimensional convolutional neural network (2D-CNN). Their performances were evaluated in terms of accuracy, sensitivity, and specificity. The result was compared with human performance. A total of four volunteers, two of whom were trained laryngologists, rated the same files. The 1D-CNN showed the highest accuracy of 85% and sensitivity and sensitivity and specificity levels of 78% and 93%. The two laryngologists achieved accuracy of 69.9% but sensitivity levels of 44%. Automated analysis of voice signals could differentiate subjects with laryngeal cancer from those of healthy subjects with higher diagnostic properties than those performed by the four volunteers.

## 1. Introduction

Laryngeal cancer is one of the most debilitating forms of malignancy, with an average incidence of 3.3 per 100,000 from 2012 to 2016 in the USA [1]. In 2019, there were 12,370 new cases diagnosed in the USA alone. Despite the rising incidence, early diagnosis remains challenging, resulting in delayed treatment [2,3]. With a delay of diagnosis, laryngeal cancer may lead to the most severe debilitating disabilities in phonation, swallowing [4] and overall quality of life. An automated voice analysis tool could advance the time of diagnosis regardless of patients’ location, in line with the idea of telemedicine. Though voice changes can indicate the first clinical signs of disease, subjective perception of early voice changes can be listener dependent and subject to intrajudge variations [5].

Image analysis based on the use of computational algorithms in radiology is now expanding to signal processing in other fields such as electrodiagnosis [6]. Furthermore, the popularity of these new techniques has led to the use of automated detection of pathological voices using machine and deep learning algorithms. A voice pathology detection was reported successful using a deep learning model [7]. Algorithms based on feature extraction, such as the Mel-frequency cepstral coefficients (MFCCs) from the acoustic signals have been used for many years to detect vocal fold disorders and dysphonia [8,9,10]. For example, Chuang et al. [9] have used normalized MFCCs features and have shown that a deep neural network (DNN) can detect abnormal voice changes in voice disorders. Another study by Fang et al. [10], which included laryngeal cancer data, have reported that the results of a DNN were superior to other machine learning algorithms.

However, in past studies, the number of cancer patients was either too small [10,11] or often assessed as a single group together with other voice disorders. Most recent studies that investigated the role of automatic detection of voice disorders [8,9,10] were based on open voice databases such as the Massachusetts Eye and Ear Infirmary Database [12] or Saarbrucken Voice Database [13], and laryngeal cancer voices were rated together as one group in combination with other voice disorders. In addition, past algorithms have not been validated against the clinicians’ judgement of voice change. Subjective perception of early voice changes can be difficult [5]. The possibility of an algorithm that can distinguish pathological voice changes at the early stages in laryngeal cancer from normal healthy voices with the potential to overcome the limitations imposed by inter-subject human perception remains to be explored.

Therefore, this study aims to investigate the role of computational algorithms including a support vector machine (SVM), extreme gradient boosting (XGBoost) and the recent popular convolutional neural network (CNN) in distinguishing voice signals of patients with laryngeal cancer against those obtained from healthy subjects. We also compared the performance levels of these algorithms to those obtained by four human raters who rated the same voice files.

## 2. Materials and Methods

### 2.1. Study Subjects

A retrospective review of medical records was performed at a single university center from July 2015 to June 2019. We identified patients who had undergone voice assessments at the time of laryngeal cancer diagnosis. Only the preoperative records were collected, whereas those obtained postoperatively or after radiotherapy were excluded.

Normal voice samples were acquired from otherwise healthy subjects who had undergone voice assessments for the evaluation of their vocal cords prior to general anesthesia for surgical procedures involving sites other than the head and neck region, such as the hands or legs. Any subject subsequently diagnosed with any benign laryngeal disease, sulcus vocalis, or one-sided vocal palsy were excluded from the data analysis of the healthy subjects. Any additional diagnosis of voice disorders was excluded by a detailed review of patients’ medical records.

Patients’ demographic information, including gender, age, and smoking history, were collected. In those diagnosed with laryngeal cancer, additional clinical information such as the TNM (Tumor Node Metastases Classification of Malignant Tumors) stage, a global standard for classifying the anatomical extent of tumor cancers, was recorded. The study protocols were approved by our institutional review board [HC19RES10098].

### 2.2. Datasets from Voice Files

The dataset comprises recordings of normal subjects and cancer patients. Voice samples were recorded with a Kay Computer Speech Lab (CSL) (Model 4150B; KayPENTAX, Lincoln Park, NJ, USA) supported by a personal computer, including a Shure-Prolog SM48 microphone with Shure digital amplifier, located at a distance of 10–15 cm from the mouth and an angle of 90°. Background noise was controlled below 45 dB. Analysis of a voice sample, directly recorded using digital technology and with a sampling frequency of 50,000 Hz, was carried out using MDVP 515 software (version 2.3). Patients phonated vowel sound /a:/ for over 4 s at a comfortable level of loudness (about 55–65 dB). The operator’s experience dates back to 2011, and the voice testing protocol in the hospital was established in 2015.

### 2.3. Experimental Setups

The study used a NVIDIA GeForce RTX2080 Ti (11GB) graphic card. We examined the normal and cancer voice signal classification and tested the performance of the SVM, XGBoost, light gradient boosted machine (LightGBM), artificial neural network (ANN), one-dimensional convolutional neural network (1D-CNN), and 2D-CNN. The performance was evaluated via five-fold cross-validation. The accuracy, sensitivity, specificity, and area under the curve (AUC) values were used as performance metrics. This study strictly used male voice samples to exclude gender effects in that all laryngeal cancer cases were male and that male and female voices have different frequency range. Otherwise, some factors that are not directly related to cancerous voice can undermine the integrity of the study design.

### 2.4. Feature Extraction

In this study, two common features in speech analysis were selected. First, a term named after the word “talk” in Dutch, PRAAT is a speech analysis program (Paul Boersma and David Weenink, Institute for Phonetic Sciences, University of Amsterdam, The Netherlands) in phonetics designed to extract key features in the voice [14]. The raw voice input was 4 s of 50,000 Hz signal. The PRAAT features were extracted under a minimum value of 10 Hz and a maximum value of 8000 Hz to account for the spectral range of a human voice. Fourteen audio features include mean and standard deviation of the fundamental frequency, harmonic to noise ratio (HNR), and jitter and shimmer variants. The HNR denotes the degree of acoustic periodicity in the aspect of energy. The last two sets of features are measures of perturbation in acoustic analysis, where jitter demonstrates the frequency instability, whereas shimmer represents the amplitude instability of a signal. In other words, jitter refers to the frequency variation from cycle to cycle, and the shimmer represents the amplitude variation of the sound wave. Following is a description of the jitter and shimmer variants [14]. The localJitter is the average absolute difference between consecutive intervals, divided by the average interval, and localabsolutejitter uses absolute difference. The rapJitter is the relative average perturbation of itself and the two adjacent, and ppq5jitter accounts for four neighbors. Ddpjitter is the difference between differences of consecutive difference. In a similar fashion, six shimmer variants were defined: localshimmer, localdbshimmer, apq3shimmer, apq5shimmer, apq11shimmer, and ddashimmer.

Second, MFCCs, a collection of numerical values resulting from the transformation of time series data, were obtained [15]. The principle of MFCCs is based on a short-time Fourier transform and additional consideration for the distinct nature of a human voice in the lower frequency range, set by the biased bandpass filter design. Forty triangular bandpass filters were used. Initially, we down sampled the input signal to 16,000 Hz, accounting for the Nyquist frequency of the human voice range. As a result of the transformation, 200,000 data points of the input signal were converted to 64,000 points, and then into a 40 × 126-time spectral image. The graphic presentation of down sampling, normalization, and MFCCs transformation is shown in Figure 1 and Figure 2. In addition, short time Fourier transform (STFT), another common time-spectral representation, was obtained for comparative evaluation. For this conversion, we down sampled the input signal to 4000 Hz and processed with a frame size of 0.02 without overlap, in order to match with the height size of the MFCCs. As a result, a 40 × 199-time spectral image was produced.

### 2.5. Preprocessing

A series of preprocessing steps are introduced for the effective representation of the signal. The recordings represent continuous sounds. Normalization was performed to change the value of numerical voice signals to a common scale, because the magnitudes vary depending on the measuring distance of the record. Each signal was divided by the maximum absolute value of the recording per patient while taking account of the peak outliers.

For accurate validation, the data set was divided into two parts in each validation: one for training and another for testing. Performance metrics were only calculated with the testing dataset, the signals that the model did not process during its training. A five-fold validation method is used for reliable results, which divides the dataset into five subsets. For each validation fold, four subsets were used to train a model for appropriate representation and generalization power, and the model was validated with the remaining subset. Overall, five validations were conducted, and the performance matrix represented the average of all results. The process can be seen in Figure 3.

### 2.6. Machine Learning Algorithms

We tested three machine learning algorithms: SVM, XGBoost, LightGBM, and ANN. The SVM is the most frequently practiced method used in the classification task. The SVM resolves the classification task by drawing a decision boundary hyperplane that divides space with the maximum distance from each class. However, not all cases can be resolved similarly, as clusters often require a non-linear boundary. The kernel trick facilitates by warping spaces. In machine learning, the hyper-parameter is a high-level configurator empirically chosen to control complexity, conservativeness, and overfitting of a model before training the networks [16]. The governing decision-making equation and its classification decision is shown in the equations below, where $\omega_{0}$ and ω represent bias and weights of the boundary.
(1)$$\omega_{0}+\;\omega^{T}x\;>\;0\;\to \;x\;\mathrm{belongs}\;\mathrm{to}\;\mathrm{normal}$$
(2)$$\;\;\omega_{0}+\;\omega^{T}x\;<\;0\;\to \;x\;\mathrm{belongs}\;\mathrm{to}\;\mathrm{pathological}$$

The LightGBM and XGBoost are classifiers derived from a decision tree family known to perform best in many practices. The decision tree is named after its shape comprising of a series of dividing rules. The model learns the optimal rules based on information gain and entropy. Information gain is a quantified value based on information generated by a certain event. Entropy is a relative degree of disorder [17]. Since a signal decision tree can easily overfit, a series of techniques are implemented to boost performance such as bagging, boosting, tree-pruning, and parallel processing. The techniques effectively combine predictions from multiple trees and multiple sequential models.

An ANN is a basic form of a DNN. A series of fully connected layers constitute an ANN. The model predicts the label of input data with trained weights and biases through a forward propagation. We consider this ANN model to be a machine learning model since the input is hand-crafted feature and the propagation mostly performs classification tasks only.

### 2.7. Deep Learning Algorithms

The human voice exhibits distinct characteristics in the lower frequency range, so biased filters are used in the MFCCs. Although a recent study has shown that MFCCs are consistent metric constraints [18], an inevitable information loss occurs at the conversion. Ten pieces of size 40 × 40 are randomly cropped per image to lower the computational cost and to elicit a data augmentation effect. In a similar fashion, ten pieces of size 40 × 40 segments from a STFT spectrogram are prepared from each signal.

Zero padding and down sampling are implemented for the 1D-CNN. Zero padding ensures stable frequency conversion and provides better resolution. Further, a recent study showed that the most contributive bands in both detection and classification ranged between 1000 and 8000 Hz [7]. Down sampling is set at 22,050 Hz for 1D-CNN and 16,000 for 2D-CNN preprocessing.

The 1D-CNN structure is composed of six convolution blocks and three fully connected layers (Figure 2). The number of kernels is 16, 32, 64, 128, and 256. A kernel is equivalent to a filter. For example, the first layer represents the filtered signals from 16 kernels. Max pooling sizes used are 8, 8, 8, 8, and 4 to compress the long signal, which choose the maximum single value from a given window size to progressively reduce the spatial size and to provide abstract representation. The dense layer is composed of 1536, 100, 50, and 2 nodes. Batch normalization and ReLU activation are used for faster and stable training [19]. The detailed structure of the algorithm is shown in Figure 4.

The 2D-CNN structure is composed of three convolution blocks and three fully connected layers. The number of kernels is 64, 64, and 128. The dense layer is composed of 500, 50, and 2 nodes. A dropout of 0.3 is used twice at a dense layer to prevent overfitting. A Glorot uniform initializer and ReLU activations are used [20]. Maximum pooling is done conservatively, only once, because the input image is already small. The detailed structure of the algorithm is shown in Figure 5.

### 2.8. Human Interpretation

Two laryngologists with 3–10 years of experience in laryngoscopy and laryngeal cancer were asked to listen to the same files and classify the voice sounds as either normal or abnormal. In addition, two volunteers with no medical background were asked to perform the same tasks. All volunteers were informed that abnormal voices are from laryngeal cancer patients, prior to the evaluation. No prior demographic information was provided. All volunteers were allowed to hear the voice files multiple times. The diagnostic parameters obtained from the four volunteers were calculated.

### 2.9. Statistical Analysis

Data are expressed as mean ± standard deviation for continuous data and as counts (%) for categorical data. Bivariate analyses were conducted using a two-tailed Student’s *t*-test for continuous data and a two-tailed χ2 or Fisher’s exact test for categorical data when appropriate.

All these statistical analyses were performed using IBM SPSS Statistics 20.0 (IBM Corp., Armonk, NY, USA), and *p*-values less than 0.05 were considered to indicate statistical significance.

Group differences between patients with cancer and healthy participants were determined using non-parametric tests. The AUC values, which reflect the diagnostic accuracy and predictive ability, were calculated for each parameter. The performance of laryngeal cancer classification was evaluated with an AUC of receiver operating characteristic (ROC) curves using roc_curve and auc functions of the Scikit-learn library and the matplotlib library in the Python 3.5.2. and R 2.15.3 package software (R Foundation for Statistical Computing, Vienna, Austria).

## 3. Results

### 3.1. Demographic Features

Using the medical records, we identified a total of 50 laryngeal cancer patients who had undergone voice analysis preoperatively. From the normal voices (*n* = 180), only the male voice data were selected (*n* = 45) and used for analysis. All cancer subjects were male. Laryngeal cancer included glottic (84%) and supraglottic (16%) types of cancer. The majority (84%) of patients were diagnosed at the T1–T2 stages when the voice recordings were performed. The characteristics of cancer and their staging are shown in Table 1. Compared with the healthy group, subjects with laryngeal cancer were significantly older, and showed higher smoking rates than the healthy subjects, as shown in Table 2.

### 3.2. Feature Selection

Figure 6 shows the contribution of each PRAAT 14 feature obtained from the XGBoost for the classification of voice changes. The HNR, standard deviation of F0, and apq11shimmer were major features in the classification of abnormal voices.

### 3.3. Accuracy of the Automatic Detection in Male Voice Samples

We performed the analysis with no female data for two reasons. First, male and female voices are known to fall within different ranges of frequency [21]. Secondly, females rarely have larynx cancer, which is directly reflected in our data set that no female data exist in cancer class. Especially in East Asian countries, the proportion of female patients with laryngeal cancer is reported to be less than 10%. Therefore, voice signals comprising only the male dataset were analyzed. Among the algorithms, the 1D-CNN again showed good accuracy levels with sensitivity levels up to 85% (Table 3). Of interest was that five out of eight supraglottic cancer patients were correctly diagnosed with the 1D-CNN model.

The accuracy values and receiver operating characteristic (ROC) curves for a set of evaluations are demonstrated in Table 3 and Figure 7. 

### 3.4. Accuracy in Human Rating

Results show large variance in the sensitivity levels across the four raters with levels as low as 29% and the highest at 50%. The two experts showed higher accuracy levels than the two non-experts, but compared to the machine learning and deep learning algorithms, they showed low sensitivity and accuracy levels. Table 4 summarizes the result.

## 4. Discussion

The results of our study provide high accuracy levels of automated algorithms using machine learning and deep learning techniques that assess voice change in laryngeal cancer. The results are promising since the majority of the cancer subjects (84%) were at early stages of cancer. Among the algorithms, the 1D-CNN showed better performance than other algorithms, with accuracy levels of 85%. All the other computational algorithms showed promising levels of performance and some showed higher accuracy levels compared with the results obtained from two laryngologists, who showed sensitivity levels of 44%. To the best of our knowledge, this is one of the first studies that has compared the performance of automated algorithms against those performed by both clinicians and non-clinicians. Based on our results, automatic detection using the 1D-CNN and other computational algorithms may be considered as potential supplementary tools to distinguish abnormal voice changes in patients with laryngeal cancer.

Past studies have already used several machine learning techniques in attempts to distinguish pathological voice disorders in laryngeal cancer. Gavidia-Ceballos and Hansen [22] demonstrated accuracy levels of 88.7% in patients with vocal fold cancer, but their sample was limited to 20 glottic cancer and 10 healthy subjects. Previous studies employed an ANN in laryngeal cancer with accuracy levels of 92% [9]. However, their data included patients who were recovering from laryngeal cancer, mostly following surgery. The voice signals in the present study were obtained from laryngeal cancer patients preoperatively, and thus, our results are more appropriate for assistance in screening laryngeal cancer rather than detection of postoperative voice changes. The results are even more promising since our study also provided detailed clinical information about laryngeal cancer, mostly at the early stages.

Our results are also in accordance with previous studies that have suggested better performance of DNNs in some datasets compared with a SVM or Gaussian mixture model (GMM) in detecting pathological voice samples [9,10,23]. An unexpected finding was that the 1D-CNN showed better performance than the 2D-CNN, a more sophisticated algorithm. The processed signals contained 64,000 (4 [s] × 16000 [Hz]) data points representing acoustic information, whereas the MFCCs carried 15,640 (40 × 391) points. In addition, the 2D-CNN model has a limited scope of 40 by 40 kernel windows at a time. Through a series of feature conversion and windowing, the 2D-CNN method leads to unfortunate information loss. Thus, the 1D-CNN is associated with a higher resolution than the 2D-CNN in the presence of appropriate hyper parameters such as learning rate, kernel size, and the number of layers. Although the 1D-CNN is more difficult to optimize, higher resolution implies higher learning capacity. Although our results are consistent with recent studies that showed the superior performance of the 1D-CNN compared with the 2D-CNN in a heart sound classification study [24], one has to be conscious of the fact that performance of these algorithms may change depending on the nature of the data.

Laryngeal cancer is known to show a skewed gender distribution [25,26] with an approximately four- to six-fold higher risk in males [27] and poor prognosis compared with females [28]. Based on this gender difference, we assessed the performance levels of these algorithms when the analysis of voice features in laryngeal cancer was limited to males. In such gender-restricted analysis, the 1D-CNN showed good performance with accuracy levels of 85.2%. This phenomenon is discrepant to those observed by Fang et al. [10], who showed that the SVM outscored the GMM and ANN when the data were analyzed separately without the female subjects. Therefore, it was unexpected that the 1D-CNN performed well even with the limited sample of male voices. Despite our results favoring the 1D-CNN, due to the uneven gender distribution in laryngeal cancer, the gender composition of the data should be considered with caution when developing future deep learning algorithms since the female voice has broader distributions in cepstral domain analysis [29]. Furthermore, one has to be mindful that the performance of these algorithms may be different depending on the feature selection and amount of data, and therefore, caution is needed before making direct comparison of which algorithm is superior to the other.

In this study, among the many PRAAT features that played significant roles in helping to classify the voice changes, of interest, the HNR was shown to be an essential feature, followed by the F0 standard deviation and apq11shimmer from the XGBoost algorithms. Past studies [30,31] have shown changes in some acoustic parameters including the HNR, jitter, and shimmer in laryngeal cancer due to the structural effects on the vibration and wave movement of the vocal cord mucosa but have not shown which parameter contributes more than the other in the classification of laryngeal cancer. Abnormal HNR values reflect an asthenic voice and dysphonia [32], which may be expected with the mass effects. Controversies surrounding the role of fundamental frequencies exist, with some suggesting these values decrease in laryngeal cancer and smokers [31], compared to healthy groups [33]. Instead, this feature is a more prominent marker in voice change observed in smokers [34]. Of interest is that no study has yet emphasized the role of the apq11shimmer values in classifying these voice changes with high accuracy. The clinical implication of the apq11shimmer value in combination with changes of the F0 standard deviation, which reflects changes in voice tone, needs to be verified with future studies, including those related to other voice disorders.

Voice changes that do not improve within two weeks are known to be one of the earliest signs of laryngeal cancer and mandate a visit to the laryngologist. Our results indicated that the average time from onset of voice change to the first visit to the laryngologist was 16 weeks. Though most patients in our study were in the early stages of cancer, in reality, patients failed to consult with the laryngologist within the initial month when the voice changes develop. Subjective perception of voice change can be challenging and our results from the ratings by the four volunteers showed that half of the early stage cases could also be missed by the human ear, even by expert laryngologists. The diagnostic parameters from the four volunteers showed overall high specificity levels, which indicate good performance levels in discerning those with normal voices. However, the low sensitivity levels indicate that human perception of subtle voice changes within the short 4 s voice file may be insufficient to discern the initial voice changes in laryngeal cancer. Though the two experts showed better performance than the two non-experts, the low sensitivity levels of this latter group are of concern and reflect real-world situations where the patients may misjudge and miss the initial changes as normal. Voice change can be the only first symptom, and if not considered as a serious sign, it can inadvertently result in a delay when making the initial visit to the laryngologist. Automated algorithms may be used to alert the “non-expert” patients when these voice changes appear to seek medical advice. Higher sensitivity levels are ideal for screening tools. Therefore, the higher sensitivity levels from the deep learning algorithms may support the use of these automated voice algorithms in the detection of voice changes in early glottic cancer. Though based on a limited number of data, our results show the potential of future applications of these algorithms in digital health care and telemedicine.

One interesting point to consider is that the 1D-CNN showed good accuracy levels, even when most were at the early stages. In addition, the inclusion of these supraglottic cancer patients who usually remain asymptomatic in the early stages and are difficult to diagnose [35] may be clinically relevant. Voice changes in advanced laryngeal cancer stages can be evident because of the involvement of the vocal cord or thyroid cartilage. By contrast, in the T1 stage, these changes may be too subtle and may go unnoticed. The encouraging results of classifying those, even in the early laryngeal cancer stages, show the opportunity of automatic voice signal analysis to be used as part of future digital health tools for the noninvasive and objective detection of subtle voice changes at the early stages of laryngeal cancer. Future studies with more detailed separate analysis among the different tumor types and stages could be promising.

Significant new work has been reported recently using artificial intelligence techniques in the early detection of cancer, including skin and gastric cancer [36,37]. Similar attempts have also been made in oropharyngeal cancer with mixed results. A few studies have used features associated with hypernasalance in oropharyngeal cancer [38] and glottal flow in glottic cancer [39] and employed ANNs with mixed results. Recent studies have shown that the CNN can be used to predict the outcome automatically [40] or detect laryngeal cancer based on laryngoscope images with accuracy levels of 86% [41], which are similar to our accuracy levels of 85.2%. Our work differs from these past studies in that we employed voice signals rather than imaging data and compared the accuracy levels of the 1D-CNN to those rated by the human ear.

The algorithms presented in this study showed promising results. However, a few limitations remain to be addressed. First is the non-inclusion of other voice disorders such as those related to more common benign disorders, such as, vocal polyps or vocal fold palsies. The main objective was to determine the accuracy of various algorithms including the 1D-CNN against those performed by human raters. The use of these algorithms to classify other voice disorders may require rebuilding the algorithm structure based on additional hyper parameters. Ongoing studies by our research group are currently attempting to design new CNN algorithms that may be used to distinguish voice changes in cancer patients from other various voice disorders, such as those related to vocal palsy or polyps. Another factor to consider is the limited number of cancer cases. Machine learning requires a large amount of processed data, and its performance depends heavily on how well the feature is crafted. The limited number of data is a problem often encountered in medical data acquired from sources other than image files. It is even more challenging to obtain voice data from patients with laryngeal cancer during the preoperative period. However, the number of cancer patients was similar to past studies [9,10,11] that employed automated algorithms in voice pathologies. The voice, a signal carrying infinite information, can be represented in a simpler form by introducing digital signal processing tools such as PRAAT or MFCCs, which improves optimization potential despite the small datasets [15]. Second, the proposed algorithms performed well for datasets comprising only males. The inclusion of females in the analysis may inadvertently provide a clue to the model with all cancer data comprising male subjects and therefore excluded female data. Although our results supported the high-performance levels of the 1D-CNN, the model proposed in this study may lose its diagnostic power when female cancer patients are included. Therefore, our algorithms require re-validation when adequate data are collected from female patients in the future. Furthermore, prospective studies are needed for large-scale validation of our model. Third, since the cancer group showed more elderly males with a higher proportion of smokers, one could question whether our algorithm classified voice changes related to the presence of laryngeal cancer or related to smoking and old age. Smoking and old age are the cardinal risk factors of laryngeal cancer. However, these two conditions manifest in distinctive voice changes. For example, according to a recent meta-analysis study [34] voice changes in smoking are manifested mostly in the fundamental frequency (F0). Likewise, voice changes in elderly males are characterized by an increase of jitter values [42]. Had our algorithms classified based solely on senile and smoking changes, these two features would have been the two most important features. Instead, other features, which may reflect the tumor effects on the voice, played a more prominent role. Nevertheless, the skewed distribution of gender, age, and smoking status are important factors to consider in future studies that intend to employ artificial intelligence in voice disorders that include laryngeal cancer. Finally, our results are by no means intended to replace current diagnostic tools and future studies using voice signals as a supplementary screening tool in the age of telemedicine in conjunction with current laryngoscope studies in laryngeal cancer are warranted.

The results presented in our study demonstrate the ability of the proposed computational algorithms to distinguish voice changes in early laryngeal cancer from healthy voices in normal participants. However, this study did not include other voice disorders, which may be more common in clinical practice than laryngeal cancer patients. Therefore, a high degree of prudence is required in interpreting the results. Nevertheless, the application of voice signals to digital algorithms as alternative methods to assess patients at difficult times [43] when direct physical contact with the laryngologist is not feasible may have important social implications in the future.

## 5. Conclusions

This study has shown that automated voice detection based on both machine learning and deep learning algorithms facilitates detection of voice changes in laryngeal cancer in a noninvasive yet objective manner with accuracy levels that may surpass human performance. Future studies are warranted on techniques to implement and adopt these automated voice analyses using the 1D-CNN, as part of the digital health system [44].

## Figures and Tables

**Figure 1 jcm-09-03415-f001:**
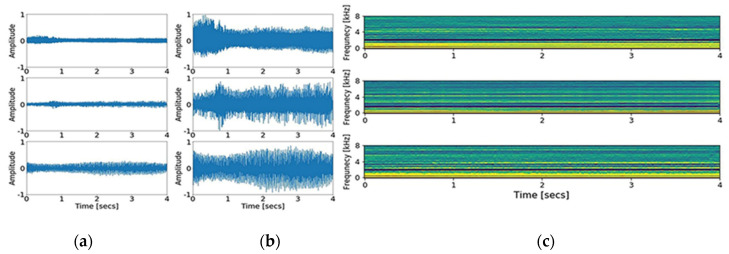
The graphic presentation of transformation from raw signal into Mel-frequency cepstral coefficients (MFCCs) image, a necessary process to comply with the two-dimensional convolutional neural network input shape. (**a**) Plot of signals down sampled to 16,000 Hz; (**b**) plot of signals normalized between −1 and 1; (**c**) image of signals after MFCCs transformation.

**Figure 2 jcm-09-03415-f002:**
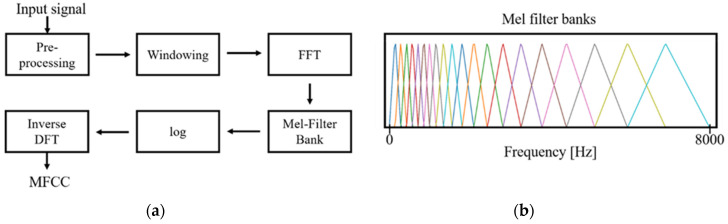
The flowchart of Mel-frequency cepstral coefficients (MFCCs) transformation. (**a**) and presentation of Mel filter banks (**b**). The triangular filter banks are densely located towards low frequency range, reflecting the distinctive nature of the human voice in that range. Abbreviations: FFT, fast Fourier transform; DFT, discrete Fourier transform.

**Figure 3 jcm-09-03415-f003:**
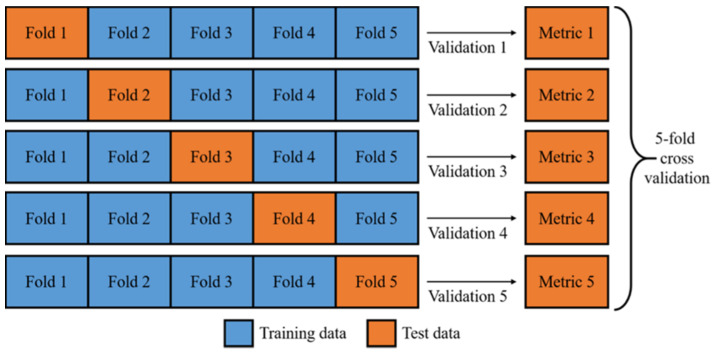
Illustration of five-fold validation. A given data set is split into five subsections where each fold is used as a testing set, a useful method to use all data where data is limited.

**Figure 4 jcm-09-03415-f004:**
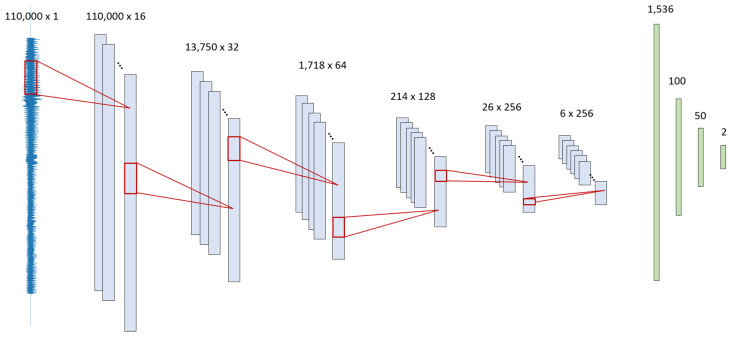
Illustration of one-dimensional convolutional neural network model structure.

**Figure 5 jcm-09-03415-f005:**
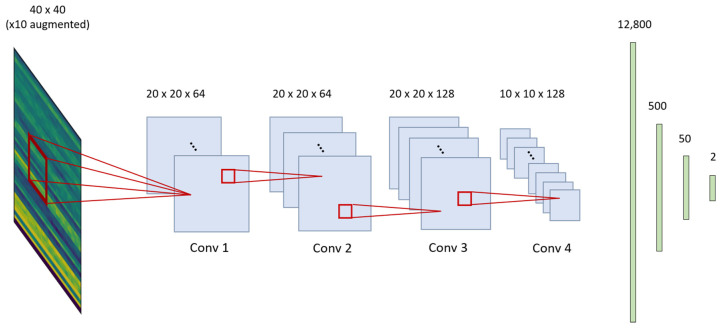
Illustration of two-dimensional convolutional neural network model structure.

**Figure 6 jcm-09-03415-f006:**
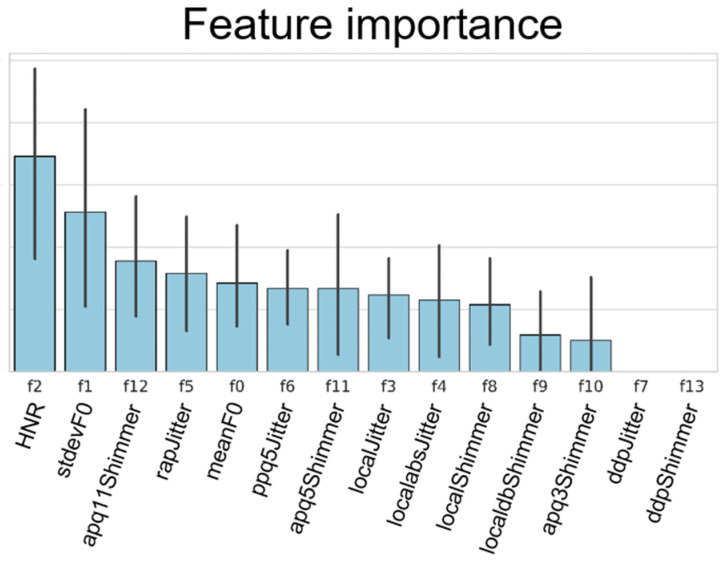
Feature importance analysis of XGBoost. The plot demonstrates the relative information gains based on the feature importance classification task of male voice samples.

**Figure 7 jcm-09-03415-f007:**
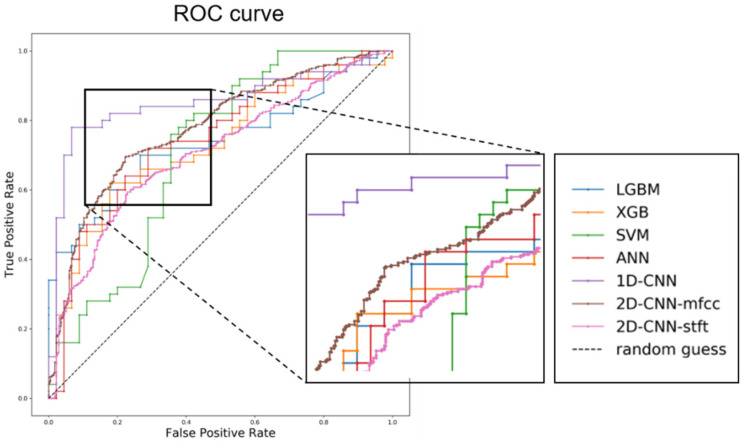
ROC (receiver operating characteristic) curve analysis of the different models for the classification of laryngeal cancer. ROC curves algorithms for classification task of only male voice samples. Abbreviations: LGBM, LightGBM; XBG, XGBoost; SVM, support vector machine; ANN, artificial neural network; 1D-CNN, one-dimensional convolutional neural network; 2D-CNN, two-dimensional convolutional neural network; MFCCs, Mel-frequency cepstral coefficients; STFT, short time Fourier transform.

**Table 1 jcm-09-03415-t001:** Clinicopathological characteristics of the cancer patients.

Characteristic		No. of Patients	[%]
Gender	Male	50	100
Primary site	Glottis	42	84
	Supraglottic	8	16
Differentiation	SCC	46	92
	Papillary SCC	2	4
	CIS	2	4
T classification	1	35	70
	2	7	14
	3	5	10
	4	3	6
N classification	0	42	84
	1	1	2
	2	4	8
	3	3	6
TNM stage	Early	39	78
	Advanced	11	22

Abbreviations: SCC; Squamous Cell Cancer, CIS; Carcinoma in Situ, TNM; Tumor Node Metastases Classification of Malignant Tumors.

**Table 2 jcm-09-03415-t002:** Demographic data from the 95 subjects, including the glottic cancer patients.

Clinical Variables		Normal Male (*n* = 45)	Larynx Cancer (*n* = 50)	*p*-Value *
Age (year)		49.7 ± 2.1 (24~83)	65.5 ± 1.3 (50~88)	<0.001
Smoking (yes)		12 (26.7)	37 (74.0)	<0.001 ^†^
Smoking amount (packs per year)	<30	7 (58.3)	8 (21.6)	
	≥30	5 (41.7)	29 (78.4)	<0.001 ^‡^

Values are presented in mean ± standard deviation (min~max) or number (%). * Group comparison between normal cases versus laryngeal cancer patients. ^†^ Hazards ratio: 14.8. ^‡^ Hazards ratio: 11.6.

**Table 3 jcm-09-03415-t003:** Evaluation metrics table of only male voice samples for classification task of abnormal voice signals in laryngeal cancer (*n* = 50) from normal healthy subjects (*n* = 45).

Algorithms	Accuracy (%)	Sensitivity (%)	Specificity (%)	AUC
SVM	70.5 (67.9–73.0)	78.0 (75.6–80.3)	62.2 (58.0–66.3)	0.708
XGBoost	70.5 (68.2–72.8)	62.0 (58.5–65.4)	80.0 (77.3–82.6)	0.731
LightGBM	71.5 (68.2–74.8)	70.0 (66.6–73.3)	73.3 (69.9–76.6)	0.739
ANN	69.4 (67.6–71.2)	62.0 (60.4–63.5)	77.7 (75.3–80.2)	0.744
1D-CNN	85.2 (83.8–86.6)	78.0 (76.0–79.9)	93.3 (92.2–94.4)	0.852
2D-CNN * (MFCCs)	73.3 (72.0–74.7)	69.6 (66.9–72.2)	77.5 (74.2–80.8)	0.778
2D-CNN * (STFT)	67.1 (65.6–68.6)	58.6 (55.6–61.5)	76.6 (75.1–78.2)	0.707

Abbreviations: AUC, area under curve; SVM, support vector machine; XGBoost, extreme gradient boosting; LightGBM, light gradient boosted machine; ANN, artificial neural network; MFCCs, Mel-frequency cepstral coefficients; STFT, short-time Fourier transform. *: with 10 times augmented data.

**Table 4 jcm-09-03415-t004:** Evaluation metrics table of only male voice samples for classification task of abnormal voice signals in laryngeal cancer (*n* = 50) from normal healthy subjects (*n* = 45).

Rater	Accuracy [%]	Sensitivity [%]	Specificity [%]	AUC
Non-expert 1	68.8 (58.3–78.2)	50.0 (35.5–64.7)	88.9 (75.9–96.3)	0.6944
Non-expert 2	56.7 (46.3–67.2)	29.1 (16.9–44.0)	86.6 (73.2–94.9)	0.5792
Expert 1	68.9 (58.7–78.0)	43.7 (29.4–58.8)	95.5 (84.8–99.4)	0.7930
Expert 2	70.9 (60.1–79.9)	43.7 (29.4–58.5)	100 (92.1–100)	0.7188
Experts Mean	69.9 (59.9–79.0)	43.7 (29.4–58.6)	97.7 (88.4–99.7)	0.7559

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
