# Peer review of "Convolutional Neural Network Classifies Pathological Voice Change in Laryngeal Cancer with High Accuracy"

_jcm, 2020, doi:10.3390/jcm9113415_

Round 1

Reviewer 1 Report

The proposed article presents whether automated voice signal analysis can distinguish patients with laryngeal cancer from healthy subjects. The five computational algorithms (SVM, XGBoost, ANN,1D-CNN, and 2D-CNN) were used for evaluating. The automated analysis of voice signals could differentiate subjects with laryngeal cancer from those of healthy subjects, and the result is higher diagnostic properties than those performed by the four volunteers.

The concept of this paper is well understandable, and the comparison of the five computer-based methods and their comparison with human performance is fascinating. I also agree with you that this method is useful for screening. However, there is a bias in the data, and there are problems with the selection of examiners and other factors that make the interpretation of the results questionable.

Some of the questions are listed below.

1. Datasets

There is quite a bit of bias associated with voice quality. The paper compared data for all genders and males only, but data for all genders are not necessary if you want to determine if it is purely a case of cancer. If you were to include the data in your paper, it would be unfair and misleading to state in the Abstract that the highest accuracy considered in voice data for all genders is 90.0%, or to base the discussion on that value.

Since it’s just a matter of determining whether you have cancer or not, you can expect to be correct 50% of the time, even if you assign it at random. If 85 female voices are identified as female, it is easy to identify them as non-cancerous. If the remaining males are then randomly assigned, the accuracy and specificity are expected over 70%. In other words, if it is easy to distinguish voices based on factors that are not related to cancer, the accuracy could be higher. Then, not only gender but also age and smoking might be an issue.

How can it be sure that it is evaluating patients with cancer? Do results show that you just identified the voice of the older man who is smoking?

Again, it is unfair and misleading to state in the Abstract the highest accuracy based on biased data or to establish the discussion on that value. At least, the data for all genders is unnecessary if you want to determine if it is purely a case of cancer.

2. Human Rating

2-a. What is the significance of being an examiner if the two volunteers have never heard the voice of laryngeal cancer? If you think these examiners are needed, do they still need to be mixed in with the otolaryngologist’s evaluation? I wonder if you want to make human-judged data look worse intentionally.

2-b. Which raters are laryngologists, and which raters are the volunteers without medical background?

2-c.

Line 348: “our results from the ratings by the four volunteers showed that half of the early stage cases could also be missed by the human ear, even by expert laryngologists.”

Line 295: “According to our knowledge, this is one of the first studies that has compared the performance of automated algorithms against those performed by clinicians.”

> The data by only laryngologists should be shown.

> Does it make sense to compare untrained humans with CNNs that have been trained repeatedly with the same data? When the volunteers evaluated, are they given at least the information that they are assessing people with cancer or normal cases with no abnormalities in their vocal cords?

2-d. Is it a coincidence that the sensitivities of Table 5 and Table 6 are exactly the same?

3. 1DCNN/2DCNN

3-a. I agree with you that 1D-CNN is better because there is a lot of information lost with 2D-CNN in evaluating voice. However, was this process of the 2D-CNN the best? Did you have a comparison with any other method of 2D-CNN?

3-b.

Line 358: One interesting point to consider was that the 1D-CNN showed good accuracy levels, even with the inclusion of these supraglottic cancer patients who usually remain asymptomatic in the early stages and are difficult-to-diagnose [33].

> I can not assess whether this is true or not because the results do not show the data separately on the glottis.

3-c.

Line 365: The promising results of classifying those even in the T1 stages,

> This paragraph talks about Supraglottic, and if you are talking about T1 in Supraglottic, the number of cases is not enough. I am afraid I have to disagree with this opinion.

3-d. Did you use data augmentation techniques on CNN? 

3-e. How much testing dataset do you use?

Author Response

Thank you for allowing us to submit a revised draft of our manuscript. We are grateful to the reviewers for their time, efforts spent, and insightful comments on our paper. We have addressed all comments provided by reviewers and revised our paper accordingly to reflect most of the suggestions.

Here is a point-by-point response to the reviewers’ comments.

We have reanalyzed to reflect the editorial suggestions for 5-fold validation after exclusion of female data. During revision, we were able to fine-tune better with some models. In addition, LightGBM model was included and 2D CNN was performed on Short Time Fourier Transform as well for comparative analysis as advised. The edited sections are highlighted in yellow for convenience.

RESPONSE TO REVIEWERS’ COMMENTS

Thank you for allowing us to submit a revised draft of our manuscript. We are grateful to the reviewers for their time, efforts spent, and insightful comments on our paper. We have addressed all comments provided by reviewers and revised our paper accordingly to reflect most of the suggestions.

Here is a point-by-point response to the reviewers’ comments.

We have reanalyzed to reflect the editorial suggestions for 5-fold validation after exclusion of female data. During revision, we were able to fine-tune better with some models. In addition, LightGBM model was included and 2D CNN was performed on Short Time Fourier Transform as well for a comparative analysis as advised. The edited sections are highlighted in yellow for convenience.

REVIEWER #1

The proposed article presents whether automated voice signal analysis can distinguish patients with laryngeal cancer from healthy subjects. The five computational algorithms (SVM, XGBoost, ANN, 1D-CNN, and 2D-CNN) were used for evaluating. The automated analysis of voice signals could differentiate subjects with laryngeal cancer form those of healthy subjects, and the result is higher diagnostic properties than those performed by the four volunteers.

The concept of this paper is well understandable, and the comparison of the five computer-based methods and their comparison with human performance is fascinating. I also agree with you that this method is useful for screening. However, there is a bias in the data, and there are problems with the selection of examiners and other factors that make the interpretation of the results questionable.

Some of the questions are listed below.

Comments:
1. Datasets

There is quite a bit of bias associated with voice quality. The paper compared data for all genders and males only, but data for all genders are not necessary if you want to determine if it is purely a case of cancer. If you were to include the data in your paper, it would be unfair and misleading to state in the Abstract that the highest accuracy considered in voice data for all genders is 90%, or to base the discussion on that value.

Since it’s just a matter of determining whether you have cancer or not, you can expect to be correct 50% of the time, even if you assign it at random. If 85 female voices are identified as female, it is easy to identify them as non-cancerous. If the remaining males are then randomly assigned, the accuracy and specificity are expected over 70%. In other words, if it is easy to distinguish voices based on factors that are not related to cancer, the accuracy could be higher.

Authors’ response: We are grateful and agree with the reviewer’s explanation on the potential bias imposed by the inclusion of the female data, which was why we initially had two sets of results: all-gender and only male. However, as accurately pointed by the reviewer, we agree that the metrics based on female data can be misleading. Hence, we have presented our data after re-analysis after the exclusion of the female data. Fortunately, the level of 1D CNN performance did not change, and we have updated the results. (please see Page 7, Line 232-235)

Also, we have commented in the limitation section that:

“Second, the proposed algorithms performed well for datasets comprising only the male. The inclusion of the female in the analysis may have inadvertently provided a clue to the model with all cancer data comprising of male subjects and therefore excluded from the normal data. Although our results supported the high-performance levels of the 1D-CNN, the model proposed in this study may lose its diagnostic power when female cancer patients are included. Therefore, our algorithms require re-validation when adequate data are collected from female patients in the future.” (please see Page 12, Line 406-412)

Then, not only gender but also age and smoking might be an issue. How can it be sure that it is evaluating patients with cancer? Do results show that you just identified the voice of the older man who is smoking?

Authors’ response: We were indeed cautious with the very same issues. Distinctive voice changes occur in the elderly male and are often manifested with an increase of jitter. Smoking per se causes inflammation and irritates the vocal cords leading to distinctive changes in the mean fundamental frequencies (F0). Therefore, had our algorithms inadvertently performed classification of an elderly male smoker, the absolute jitter and the mean fundamental frequencies would have been expected to show higher importance. Theoretically, the influence of a laryngeal mass would be expected to manifest in different acoustic features, and indeed the three most important features were the HNR, standard deviation of F0 and apq 11 shimmer. Based on our findings, one could cautiously suggest that our algorithms may have discerned voice changes related to glottic mass and not necessarily to those solely related to gender or smoking effects. However, in real-world situations, smoking and increased age are cardinal risk factors for laryngeal cancer and would be difficult to eliminate their effects. The potential of these two clinical factors to cause potential bias the data were addressed in the limitation section as follows:

“Third, since the cancer group showed more elderly males with a higher proportion of smokers, one could question whether our algorithm classified voice changes related to the presence of laryngeal cancer or related to smoking and old age. Smoking and old age are the cardinal risk factors of laryngeal cancer. However, these two conditions manifest in distinctive voice changes. For example, according to a recent meta-analysis study voice changes in smoking are manifested mostly in the fundamental frequency (F0). Likewise, voice changes in the elderly male are characterized by an increase of jitter values. Had our algorithms classified based solely on senile and smoking changes, these two features would have been the two most important features. Instead, other features, which may reflect the tumor effects on the voice, played a more prominent role. Nevertheless, the skewed distribution of gender, age and smoking status are important factors to consider in future studies that intend to employ artificial intelligence in voice disorders that include laryngeal cancer.” (please see Page 12, Line 413-424)

Again, it is unfair and misleading to state in the Abstract the highest accuracy based on biased data or to establish the discussion on that value. At least, the data for all gender is unnecessary if you want to determine if it is purely a case of cancer.

Authors’ response: Thank you for pointing out the bias in the data. We are grateful and agree with the reviewer’s explanation on the inclusion of the female data, which was why we initially had two sets of results: all-gender and only-male. However, as accurately pointed by the reviewer, we agree the metrics based on female data can be misleading. Hence, we have presented our re-analyzed result after the exclusion of the female and have updated our paper accordingly.

Comments:
2. Human Rating

2-a What is the significance of being an examiner if the two volunteers have never heard the voice of laryngeal cancer? If you think these examiners are needed, do they still need to be mixed in with the otolaryngologist’s evaluation? I wonder if you want to make human-judged data look worse intentionally.

Authors’ response: Thank you for your comment. We agree with the reviewer’s advice that two volunteers’; the non-experts; evaluations need to be separated with the laryngologists. We kept the accuracy of these two non-experts, considering them as a base-line, but separated the evaluation metrics of experts as advised. (please see Page 10, Line 285-287, Table 4). As for the justification of including non-experts, we definitely had no intentions to present human judgement worse. Instead we wanted to show that subtle voice changes can be difficult to perceive especially in those with no medical background. We have commented that:

“Though the two experts showed better performance than the two non-experts, the low sensitivity levels of this latter group are of concern and reflect real-world situations where the patients may misjudge and miss the initial changes as normal. Voice change can be the only first symptom, and if not considered as a serious sign, it can inadvertently result in a delay when making the initial visit to the laryngologist. Automated algorithms may be used to alert the “non-expert” patients when these voice changes appear to seek for medical advice. Higher sensitivity levels are ideal for screening tools. Therefore, the higher sensitivity levels from the deep learning algorithms may support the use of these automated voice algorithms in the detection of voice changes in early glottic cancer. Though based on a limited number of data, our results show the potential of future application of these algorithms in digital healthcare and telemedicine.” (please see Page 11, Line 361-370)

Also, the voice data sets provided consisted of a ‘vowel sound /a:/ for over 4 seconds at a comfortable level of loudness (about 55-65 dB)’. This short duration of voice could have limited the expert’s interpretation of voice change. In real-world situations, the performance would have exceeded than those shown in our study since laryngologists have more time to listen to the voice and talk to the patient. The aim of this study, as stated in the limitations, was not intended to replace current diagnostic tools or the role of clinicians. However, we feel that with the advent of telemedicine, especially during these difficult CoVid19 times, the usage of such technologies could potentially help both laryngologists and patients when direct physical contact is not deemed feasible.

We have added that:

“However, the low sensitivity levels indicate that human perception of subtle voice changes within the short 4 second voice file, may be insufficient to discern the initial voice changes in laryngeal cancer.” (please see Page11, Line 360-361)

2-b. Which raters are laryngologists, and which raters are the volunteers without medical background?

Authors’ response: Thank you for your comment. We have identified the 4 volunteers accordingly on Table 4 to clear the confusion. (please see Page 10, Line 285-287, Table 4)

2-c.

Line 348: “our results from the ratings by the four volunteers showed that half of the early stage cases could also be missed by the human ear, even by expert laryngologists.”

Line 295: “According to our knowledge, this is one of the first studies that has compared the performance of automated algorithms against those performed by clinicians.”

> The data by only laryngologists should be shown.

> Does it make sense to compare untrained humans with CNNs that have been trained repeatedly with the same data? When the volunteers evaluated, are they given at least the information that they are assessing people with cancer or normal cases with no abnormalities in their vocal cords?

Authors’ response: We appreciate your valuable comments on this matter. We agree with the reviewer’s comment that the results of untrained humans do not carry significance because they have never heard of cancer patients’ voice. We meant to provide the result of non-expert for the purpose of setting a base-line and separated them with the evaluation values of experts as advised by the reviewer. Further, we have split the dataset into train and test set so that CNNs always evaluate on the unseen test data. Everyone was informed that abnormal voices are from laryngeal cancer patients, prior to the evaluation. As stated earlier, we have identified the non-experts and experts in the manuscript. We have included this information in the text as it is critical. (please see Page 6, Line 213-216) We have also corrected that this study “…has compared the performance of automated algorithms against those performed by both clinicians and non-clinicians.” (Page 10, Line 295-297)

2-d. is it a coincidence that the sensitivities of Table 5 and Table 6 are exactly the same?

Authors’ response: Thank you for pointing this out. The numbers of the actual cancer patients were all male and the correct number of true positives stayed the same for both Tables 5 and 6. In the revised manuscript only Table 6 (now Table 4, Page 10, Line 285-287) is to be presented.  

Comments:
3. 1DCNN/2DCNN

3-a.

I agree with you that 1D-CNN is better because there is a lot of information lost with 2D-CNN in evaluating voice. However, was this process of the 2D-CNN the best? Did you have a comparison with any other method of 2D-CNN?

Authors’ response: Thank you for providing good comment. This was the best performance of many 2D-CNN architectures and hyper-parameter searching we have implemented in this study. Definitely, a better result can be made with some modification with structure such as adaptation of inception module or better fine-tuned values. A 5-fold validation was redone to reflect the editor’s suggestion and with further fine-tuning, the accuracy is slightly increased to 73% in the case of 2D-CNN. With regards to the conversion method in the pre-processing, we included the short-time Fourier transform for the comparative evaluation. (please see Page 3, Line 131-135; Page 9, Line 266-271, Table 3)

3-b.

Line 358: One interesting point to consider was that the 1D-CNN showed good accuracy levels, even with the inclusion of these supraglottic cancer patients who usually remain asymptomatic in the early stages and are difficult-to-diagnose [33 (now 35)].

> I cannot assess whether this is true or not because the results do not show the data separately on the glottis.

Authors’ response: Thank you for pointing this out. We agree with reviewers comment that there is a need for more supporting material. Five out of eight supraglottic cancer patients were correctly diagnosed with 1D CNN model, and this was pointed in the results section. (please see Page 8, Lines 261-262).

However, with the limited number of supraglottic patients in this study, making such broad statements could be misleading, and we have, therefore, changed the paragraph (as shown in the comment below).

3-c.

Line 365: The promising results of classifying those even in the T1 stages,

> This paragraph talks about Supraglottic, and if you are talking about T1 in Supraglottic, the number of cases is not enough. I am afraid I have to disagree with this opinion.

Authors’ response: Thank you for your valuable comment. We agree that this paragraph can be misleading, and we have therefore made the edited the sentences so that the revised paragraph states:

“one interesting point to consider was that the 1D-CNN showed good accuracy levels, even when most were at the early stages. Also, the inclusion of these supraglottic cancer patients who usually remain asymptomatic in the early stages and are difficult-to-diagnose [35] may be clinically relevant. Voice changes in advanced laryngeal cancer stages can be evident because of the involvement of the vocal cord or thyroid cartilage. By contrast, in the T1 stage, these changes may be too subtle and may go unnoticed. The encouraging results of classifying those even in the early laryngeal cancer stages, show the opportunity of automatic voice signal analysis to be used as part of future digital health tools for the noninvasive and objective detection of subtle voice changes at the early stages of laryngeal cancer. Future studies with more detailed separate analysis among the different tumor types and stages could be promising.” (please see Page 11-12, Line 371-380)

3-d. Did you use data augmentation techniques on CNN?

Authors’ response: Thank you for the comment. For 2D CNN, each signal is augmented by 10 times during random cropping process from MFCC or Short Time Fourier Transform images. (please see Page 5, Lines 187-189).

3-e. How much testing dataset do you use?

Authors’ response: Thank you for the comment. We use 5 fold validation method. The train/test split is 80/20. Therefore, in each fold, 76 samples (36 normal, 40 cancer) are used for training and the rest unseen 19 (9 normal, 10 cancer) are used for testing. In total of 5 independent, separate fold evaluation, the whole sample data (95 samples) are used for testing. (please see Page 5, Lines 159-161, Figure 3)

We appreciate your feedback in our work and hope that the revisions we have prepared addresses all your concerns and the revised manuscript is deemed suitable for publication.

Thank you for your consideration.

Reviewer 2 Report

This article present an experiment in order to detect the presence of a cancer based on recordings of the voice of patients.

Different methods of machine learning are tested: SVM, XGBoost, ANN and deep learning ones: CNN (1D and 2D).

The experiments are very well presented, features are well describe, process is well detailed. Identified limitations of the study are explained (the non-inclusion of other voice disorders, the biais with female patients) and it will be interesting to assess this in the future.

I am embarrassed by the fact that the paper seems to oppose the methods of classification and draw definite conclusions. The results of the classification methods can vary because of: feature used, corpus, amount of data, topology of the system, features of the system, initialization... The presentation of 1D CNN here performs well: it is interesting, but this can not disqualify the orders methods on other data and experiment. The topology used for your system is adequate for your problem, and maybe alterations could provide better or lesser performances.

Assertion like (l. 309): "Our results are also in accordance with previous studies that demonstrated the superior performance of DNN compared with SVM or Gaussian Mixture Model (GMM) in detecting pathological voice samples [9,10,23]". These studies show that DNN in their studies performed better than SVM ou GMM. I think that depends of the amount of data available. DNN can train more complex models, but with sufficient data for the learning.

Author Response

RESPONSE TO REVIEWERS’ COMMENTS

Thank you for allowing us to submit a revised draft of our manuscript. We are grateful to the reviewer's time, efforts spent, and insightful comments on our paper. We have addressed all comments provided by reviewers and revised our paper accordingly to reflect most of the suggestions.

Here is a point-by-point response to the reviewers’ comments.

REVIEWER #2

This article present an experiment in order to detect the presence of a cancer based on recordings of the voice of patients.

Different methods of machine learning are tested: SVM, XGBoost, ANN and deep learning ones: CNN (1D and 2D).

The experiments are very well presented, features are well described, process is well detailed. Identified limitations of the study are explained (the non-inclusion of other voice disorders, the bias with female patients) and it will be interesting to assess this in the future.

Comments:
I am embarrassed by the fact that the paper seems to oppose the methods of classification and draw definite conclusions. The results of the classification methods can vary because of: feature used, corpus, amount of data, topology of the system, features of the system, initialization… The presentation of 1D CNN here performed well: it is interesting, but this cannot disqualify the orders methods on other data and experiment. The topology used for your system is adequate for your problem, and maybe alterations could provide better or lesser performances.

Authors’ response: Thank you for your valuable comment. To avoid this misunderstanding in the revised manuscript we were more cautious with the words and terms selected to prevent such unintentional message. We agree with the reviewer’s comment that the performance of classification method can vary depends on many factors. For example, in this particular study we only have used 14 features of which 2 were found meaningless from feature importance analysis. If we used more informative, definitive features, ML algorithms could easily exceed other algorithms. We had no intent to disqualify the algorithms performed less in this study. To further stress that some ML techniques can show promising results, we included one additional ML algorithm, LightGBM which is a decision tree family, during revision.

Comments:
Assertion like (l. 309): “Our results are also in accordance with previous studies that demonstrated the superior performance of DNN compared with SVM or Gaussian Mixture Model (GMM) in detecting pathological voice samples [9, 10, 23]”. These studies show that DNN in their studies performed better than SVM or GMM. I think that depends of the amount of data available. DNN can train more complex models, but with sufficient data for the learning.

Authors’ response: We appreciate your valuable comment. We agree with the reviewer’s comment that DNN has capacity to learn more complex relations by design, but capacity means more parameters to tune and to sufficient data for the learning. In light with the comments we have made above, we have added that:

“one has to be conscious of the fact that performance of these algorithms may change depending on the nature of the data,” (please see Page 10, Lines 321-322)

and

“one has to be mindful that the performance of these algorithms may be different depending on the feature selection and amount of data and therefore caution is needed before making direct comparison of which algorithm is superior to the other.” (please see Page 11, Lines 333-336)

We appreciate your enthusiasm in our work and hope that the revisions we have prepared address all the concerns and the revised manuscript is deemed suitable for publication.

Thank you for your consideration. We look forward to hearing from you.

Round 2

Reviewer 1 Report

I greatly appreciate your sincere reply. All my questions have been carefully answered, and I think the paper has been fascinating and meaningful.
I wish you and your colleagues continued success in your research.